# The Effect of Carbon Content on the Microstructure and Mechanical Properties of Cemented Carbides with a CoNiFeCr High Entropy Alloy Binder

**DOI:** 10.3390/ma15165780

**Published:** 2022-08-21

**Authors:** Cheng Qian, Yong Liu, Huichao Cheng, Kun Li, Bin Liu, Xin Zhang

**Affiliations:** State Key Laboratory of Powder Metallurgy, Central South University, Changsha 410083, China

**Keywords:** cemented carbides, microstructure, transverse rupture strength, fracture toughness, two-phase region

## Abstract

CoNiFeCr high entropy alloy (HEA) was used as a binder in cemented carbides for developing a new high-performance binder. The microstructure of WC-HEA cemented carbides with different binder components and carbon contents was subsequently studied. It was observed that the (Cr,W)C phase precipitated at the WC/HEA interface, and a coherent interface with a low degree of misfit was formed between WC and (Cr,W)C, thereby resulting in a significant reduction in the interfacial energy and stress concentration. The (Cr,W)C phase exerted a pinning force (Zener-drag) on the moving grain boundaries, which effectively inhibited the growth of WC grains. As a result, compared with WC-Co, WC-CoNiFeCr had smaller WC grain size, smoother grain shape and larger mean free path (MFP) of the binder, which resulted in slightly lower hardness and higher transverse rupture strength (TRS) and fracture toughness. The lower limit of carbon content in WC-CoNiFeCr was higher than that of WC-Co. With the addition of Ni, the width of the two-phase region became wider, whereas the width of the two-phase region became narrower with the addition of Fe and Cr.

## 1. Introduction

The conventional cemented carbide possesses the refractory carbide matrix (e.g., WC, TiC and TaC), with ductile Co as the binder. Due to its excellent bending strength, wettability and work hardening properties, Co has been widely used in producing WC-Co cemented carbides [1,2,3,4]. However, as a key and scarce strategic metal, Co is expensive, and its toxicity and pollution issues cannot be ignored. At the same time, in actual working conditions, poor toughness has always been a thorny issue limiting the development of cemented carbides. In addition to maintaining excellent hardness, optimal strength and toughness can significantly extend the service life of the cemented carbides [5,6,7,8,9,10,11,12]. Therefore, low-cost, safe and environmentally friendly alloys with high toughness have become the focus of attention in the cemented carbide industry [13,14,15,16].

High entropy alloys (HEAs) are multi-component alloys containing an almost equivalent percentage (5−35%) of at least four metallic elements [17,18]. Due to their high strength, toughness and corrosion resistance, HEAs have been extensively applied as binders in the cemented carbides in recent years [7,19,20,21,22].

Due to its high compatibility with Co, Cr represents one of the important elements in HEA binders. However, HEA binders usually contain up to 25% Cr, which remarkably narrows down the carbon window of the cemented carbides, along with reducing the roofing of the two-phase area and enhancing the sintering difficulties, resulting in irregular grain shapes and poor toughness. For instance, Raquel et al. [23] studied the phase formation in WC-FeCoNiCrCuAl_0.5_ and reported that a significant amount of brittle Cr-rich carbides was present in the microstructure, forming an interconnected carbide network. As a consequence, the composition of the metallic binder after sintering changed dramatically, and the amount of the metallic binder exhibited a significant decline. Dong et al. [24] reported that by replacing Co with CrMnFeCoNi as the binder, the η-carbide phase inevitably appeared in the hard metal, due to the reaction between the HEA elements and WC during liquid phase sintering. Holmström et al. [17] determined the phase diagram of the WC-CoCrFeNi multi-component system using existing databases and observed that the carbon window of the HEA binder composite was shifted about 0.2 wt.% towards higher carbon content in relation to the Co reference. In addition, the Cr-rich M_7_C_3_ phase (M → Cr, Co, W) appeared in the HEA alloy composite.

The appearance of the brittle M_7_C_3_ phase leads to a remarkable reduction in the fracture toughness of the cemented carbides. Therefore, based on the concept of HEAs, a multi-component binder with a non-equal atomic ratio was adopted in this study. WC-CoNiFeCr with different carbon contents was used, while WC-Co was used for comparison. The Cr/(Co+Ni+Fe+Cr) ratio was kept constant as 0.05 in all compositions to avoid the precipitation of the Cr-rich M_7_C_3_ carbides [25]. The mechanical properties and microstructure of the cemented carbides were studied as a function of the carbon content. Finally, the influence of Ni, Fe and Cr elements on the movement of the carbon window as well as the lower limit of the carbon content for WC-CoNiFeCr were analyzed. This study not only provides a scientific basis for the preparation of cemented carbides with high TRS and fracture toughness, but also expands a new approach for the application of HEA in the field of cemented carbides.

## 2. Experimental

### 2.1. Material Preparation

High purity WC (2.0 μm), Co (1.52 μm), Ni (2.5 μm), Fe (2.65 μm), Cr_2_C_3_ (3.0 μm) and W (1.0 μm) powders were chosen as starting materials to fabricate WC-10Co-8Ni-1Fe-1Cr (wt.%) and WC-10Co-7Ni-2Fe-1Cr (wt.%) (WC-CoNiFeCr) materials with different carbon contents by employing sintering-hot isostatic pressing (SHIP), which is adding a low pressure during sintering to eliminate defects, reduce porosity and improve comprehensive performances of cemented carbides. The WC powders were provided by Starck Corporation in Shanghai, China, whereas the Co, Ni, Fe, Cr_2_C_3_ and W powders were procured from GEM Corporation in Shenzhen, China. WC-20Co (wt.%) (WC-Co) was also prepared using the same conditions to compare the mechanical properties of the cemented carbides with the CoNiFeCr binder and traditional material. For convenience, WC-10Co-8Ni-1Fe-1Cr samples with different carbon contents were denoted as 1, 2, 3, 4 and 5, and the WC-10Co-7Ni-2Fe-1Cr samples with different carbon contents were denoted as A, B, C, D and E. The nominal compositions of WC-CoNiFeCr and WC-Co are listed in Table 1. The mixed powders were milled for 36 h in alcohol in an Ar atmosphere. The ball-to-powder weight ratio was 4:1, and the rotational speed of the mill was 35 rpm. Subsequently, the powders were mixed with 2 wt.% paraffin, followed by granulation, sieving and compression into rectangular specimens. The consolidation was performed in a SHIP furnace at 1420 °C for 1 h in an Ar atmosphere at a pressure of 5 MPa.

### 2.2. Mechanical Properties

The hardness was measured by employing an HV-Vickers hardness tester using a load of 30 kgf and a dwell time of 10 s. The micro-hardness of the binder was measured using a nanoindenter (UNHT+MCT+MST).

Based on the standard GB/T 3851-2015, the transverse rupture strength (TRS) was determined with a type B specimen by using a three-point bending tester.

As per the GB/T 4161-2007 standard, the fracture toughness was measured with a single-edge notched bending (SENB) specimen:(1)$$B:W:S=1:2:8,\;L>4.2W,0.45\leq \frac{a}{W}\leq 0.55$$
where thickness (*B*) is 8 mm, width (*W*) is 16 mm, span (*S*) is 64 mm and length (*L*) is 70 mm.

The fracture toughness, *K_I_**_c_*, is determined as follows:(2)$$K_{Ic}=\frac{P}{B\sqrt{W}}f(\frac{a}{W})$$
where *B* is the sample thickness (cm), *W* is the sample width (cm), *P* is the maximum load during the fracture (kN), *a* is the final crack length (cm) and *f*(*a*/*W*) is a dimensionless function related to *a*/*W*. For the SENB sample, *f*(*a*/*W*) can be expressed as:(3)$$f(\frac{a}{W})=\frac{3\frac{S}{W}\sqrt{\frac{a}{W}}}{2(1+2\frac{a}{W}){\left(1-\frac{a}{W}\right)}^{\frac{3}{2}}}[1.99-\frac{a}{W}(1-\frac{a}{W})\{2.15-3.93(\frac{a}{W})+2.7{\left(\frac{a}{W}\right)}^{2}\}]$$

### 2.3. Microstructural Characterization

The microstructural observation was performed using a scanning electron microscope (SEM, Quanta FEG 250, FEI Company in Hillsboro, OH, USA) equipped with an energy dispersive X-ray (EDX) spectrometer. The crystal phases were identified by using an X-ray diffractometer (XRD, D/MAX-2250, Rigaku Corporation in Tokyo, Japan) with CuKα radiation. The elemental distribution was determined using an electron probe microanalyzer (EPMA, JXA–8230, JEOL in Tokyo, Japan). The EBSD analysis was performed by using a dual-beam SEM (Helios NanoLab G3 UC, FEI Company in Hillsboro, OH, USA), equipped with an orientation imaging microscopy system and OIM analysis software (OIM 7). The thin slices, prepared using argon ion milling, were characterized by HREM, using a Talos F200S microscope.

## 3. Results

### 3.1. Microstructure

Figure 1 presents the XRD patterns of the WC-CoNiFeCr cemented carbides. As can be observed, the Cr-rich M_7_C_3_ phase does not appear. Except for Alloys 4, 5 and E, composed of the WC and binder, the other cemented carbides are composed of the WC phase, binder and η phase. The η phase is noted to be of M_6_C type (M → Co, Ni, Fe, Cr, W). Moreover, as the carbon content increases, the peak intensity of the η phase weakens (i.e., the content of the η phase decreases). It can also be observed that the binder (CoNiFeCr) has a face-centered cubic (FCC) structure, with the three main peaks positioned from left to right as (111), (200) and (220). However, the 2θ values are different, and the lattice constants of 10Co-8Ni-1Fe-1Cr and 10Co-7Ni-2Fe- 1Cr are observed to be 3.583 Å and 3.578 Å, while the constant of Co is 3.554 Å, demonstrating that the peaks of the CoNiFeCr phase undergo a change as compared with the Co phase, due to the solid solution effects.

Figure 2 presents the microstructure of WC-10Co-8Ni-1Fe-1Cr and WC-Co. For the carbon content ranging between 4.5% and 4.7% (Alloy 1–Alloy 3), the cemented carbide is observed to be composed of WC, binder and η phase (M_6_C). The light gray phase in the faceted shapes (triangles or quadrangles) is noted to be the WC phase. The black phase is the binder, whereas the dark gray phase in irregular or rounded shapes represents the η phase. The microstructure is observed to be dominated by WC grains (1–2 μm), around which the binder is noted to be evenly distributed. A small fraction of binder is locally enriched, thereby forming numerous dispersed “binder pools” (2–5 μm). At a low carbon content, the segregation of the η phase (1–6 μm) is observed to be more serious. However, as the carbon content increases, the η phase decreases, whereas the binder phase increases. In addition, the distribution of the η phase becomes dispersed and uniform. For the carbon content ranging between 4.8% and 4.9% (Alloys 4 and 5), the η phase disappears, and the cemented carbide is composed of WC and binder. At the same time, the binder pool is hard to observe. Comparing Figure 2d–f, it can be detected that the grain size of Alloys 4 and 5 is significantly smaller compared to WC-Co, and the grain shape is less sharp.

The EDS analysis in Figure 3 reveals that Alloy 1 is composed of WC, binder and η phase. Four alloying elements (Co, Ni, Fe and Cr) can be considered to be essentially homogeneously distributed in the binder, and the content of Co and Ni is relatively high, while the content of Fe and Cr is relatively low. Thus, it can be stated that the binder is composed of CoNiFeCr.

Figure 4 illustrates the microstructure of WC-10Co-7Ni-2Fe-1Cr. For the carbon content range 4.5–4.8% (Alloy A–Alloy D), the cemented carbide is observed to be composed of WC, binder and η phase. Similarly, as the carbon content increases, the η phase and binder pool decrease. At a carbon content of 4.9% (Alloy E), the cemented carbide is composed of WC and binder. The cluster observed in Alloy C (Figure 4c) was introduced in a previous study [26].

The binder volume fraction (V_binder_), the volume fraction of η phase (V_η_) and the contiguity of the carbide phase (C_WC_) of WC-CoNiFeCr and WC-Co, measured by ImageJ software from the SEM images (ImageJ 1.48), are listed in Table 1. It can be found that when the carbon content is less than the lower limit, the generated η phase leads to an increase in V_η_ and a decrease in V_binder_ and C_WC_. When the carbon content is in the two-phase region, the V_binder_ and C_WC_ values of WC-HEA and WC-Co are similar.

### 3.2. Mechanical Properties

The mechanical properties of WC-CoNiFeCr and WC-Co cemented carbides are presented in Figure 5. Figure 5a illustrates the hardness of the WC-CoNiFeCr cemented carbides as a function of the carbon content. The hardness is noted to be inversely proportional to the carbon content and decreases upon increasing the carbon content. In the two-phase region (no η phase), the hardness of the WC-Co cemented carbide (HV 966) is higher than that of WC-CoNiFeCr. Moreover, the hardness of WC-10Co-7Ni-2Fe-1Cr (HV 951) is observed to be higher than that of WC-10Co-8Ni-1Fe-1Cr (HV 935).

Figure 5b presents the TRS of the WC-CoNiFeCr cemented carbides as a function of the carbon content. As can be observed, the TRS increases with the carbon content. In the two-phase zone (no η phase), the TRS of the WC-Co cemented carbide (3146 MPa) is lower than that of WC-CoNiFeCr. Moreover, the TRS of WC-10Co-8Ni-1Fe-1Cr (3803 MPa) is higher than that of WC-10Co-7Ni-2Fe-1Cr (3543 MPa).

Figure 5c presents the fracture toughness of Alloy 5, Alloy E and WC-Co. As can be observed, the fracture toughness of the samples follows the sequence: Alloy 5 (21.26 MPam^1/2^) > Alloy E (20.18 MPam^1/2^) > WC-Co (19.30 MPam^1/2^).

## 4. Discussion

### 4.1. Evolution of Two-Phase Region

Figure 6 illustrates the thermodynamic calculations of a vertical section of the phase diagram in the WC-20Co cemented carbide. The red lines indicate the phase boundary of the η phase, the blue lines denote the phase boundary of graphite and the black lines represent the phase boundaries of liquidus and solidus. The light blue, yellow and gray shaded areas correspond to the width of the η phase area, carbon window and free carbon area, respectively. As can be observed, the width of the carbon window of WC-20Co is 4.55–4.97%.

In the case where the carbon content is below the lower limit, the WC, binder and W react to generate the η phase. The η phase generated during the liquid phase sintering process increases with a reduction in the carbon content, as a part of the binder needs to be consumed to generate the η phase, thereby reducing the binder content. Moreover, the η phase is mainly formed in the original binder. Upon increasing the content of the η phase, the segregation phenomenon of the η phase occurs due to the migration of binder [27,28]. In the case where the carbon content is higher than the lower limit, only the WC and binder are present in the alloy, and the η phase disappears.

At a carbon content of 4.7%, both Alloy 3 and C possess the η phase. At a carbon content of 4.8%, Alloy 4 is already in the two-phase region; however, Alloy D still has the η phase. In the case where the carbon content is 4.9%, Alloys 5 and E are both in the two-phase region, indicating that the lower limit of the carbon content in WC-10Co-8Ni-1Fe-1Cr is between 4.7% and 4.8%, whereas the lower limit of the carbon content in WC-10Co-7Ni-2Fe-1Cr lies between 4.8% and 4.9%; both are higher than the lower limit of the carbon content in WC-20Co (4.55%). Moreover, with the addition of Cr and Fe elements, the lower limit of the carbon content is noted to shift upwards, and the width of the two-phase region becomes narrower. In contrast, with the addition of Ni, the lower limit of the carbon content is observed to shift down, and the width of the two-phase region increases [28,29,30].

### 4.2. Microstructural Evolution: WC/HEA Interface

Figure 7 demonstrates the TEM analysis of Alloy 5. The TEM bright field image (Figure 7a) illustrates that the morphology of the WC grains is related to the P-6m2 hexagonal crystal structure, delimited by the basal {0001} and prismatic {10-10} planes. A fraction of WC grains are of triangular prism shape, with habit planes and sharp corners. The other WC grains are of triangular prism shape; however, the edges are truncated. Moreover, the size of WC grains with truncated edges is much finer than those with sharp edges, indicating that the growth of the round WC grains is suppressed. Figure 7b,c represent the numerous steps observed at the WC/HEA interface. The HREM image of the WC/HEA interface observed along the <0001> zone axis is provided in Figure 7d. A layer, about 3–4 atom planes thick, is observed at the interface between WC and HEA. The precipitate is observed to be a typical face cubic centered (FCC) crystal along the <-1-10> zone axis. The interplanar spacing d is close to 0.2912 nm, whereas the lattice parameter a is close to 0.4118 nm. Table 2 illustrates the EDS composition (at.%) in the binder and at the interface in the yellow rectangles in Figure 7d. The Cr/Co+Ni+Fe+Cr ratio measured at the interface is closely dependent on the position of the probe, with respect to the WC and binder. The Cr/Co+Ni+Fe+Cr ratio is higher at the interface (0.187) than the binder (0.043), indicating Cr interfacial enrichment. Moreover, the content of W and C is higher at the interface than the binder. Combining the crystal structure and compositions, previous studies [9,31] have reported that the (Cr,W)C phase precipitates at the interface in the WC-Co-Cr cemented carbide. The (Cr,W)C phase is an FCC structure of NaCl type, and the lattice parameter of (Cr,W)C is slightly larger than that of CrC (0.403 nm), due to the possible replacement of Cr atoms by larger W atoms. The lattice parameters a and c of the WC are determined to be 0.2906 nm and 0.2838 nm, respectively, and the misfit values (d_2_ − d_1_/d_1_) along <0001>_WC_//<-1-10>_(Cr,W)C_ are 0.2% in the basal plane (d_(Cr,W)C_ = 0.2912 nm and d_WC_ = 0.2906 nm) and 2.6% in the prismatic plane (d_(Cr,W)C_ = 0.2912 nm and d_WC_ = 0.2838 nm). As the misfit values are less than 5%, the interface between WC and (Cr,W)C is observed to be coherent. The coherent interface with a low degree of misfit results in a significant reduction in the interface energy and stress concentration. The fast Fourier transformation (FFT) patterns (Figure 7e,f) reveal that the HCP-WC binder and FCC-HEA are preserved as main constituents in the WC-HEA cemented carbide.

### 4.3. Microstructural Evolution: WC Grains and HEA Binder

Table 3 illustrates the composition of the binder (wt.%) in different samples measured by point scanning with an EPMA. For the same binder, the binder composition in the samples with different carbon contents is basically the same. More W is dissolved in the binder in cases where the η phase is present in the sample, while the solubility of W in the binder decreases in cases where the sample is in the two-phase region. In the cooling stage of the sintering process, the η phase decomposes into WC and binder. Therefore, in the case where the η phase is present, the solubility of W in the binder is enhanced. Compared with WC-10Co-8Ni-1Fe-1Cr, the composition of the binder WC-10Co-7Ni-2Fe-1Cr exhibits an increase in the proportion of the Fe element, while the proportion of the Ni element decreases. In the case where the sample is in the two-phase region, the solubility of W in Co (39 wt.%) is higher than that of CoNiFeCr (22–24 wt.%). Therefore, as compared to Co, the CoNiFeCr HEA binder exhibits a lower solubility for WC.

Figure 8 provides the fracture morphology of Alloy 5, Alloy E and WC-Co. In WC-Co, the fracture type is basically an intergranular fracture (C/C), with a smooth surface along the WC/WC grain boundary and the C/B fracture extending along the WC/binder interface. In Alloy E, except for the C/C and C/B fractures, the transgranular fracture (C) through the WC grains is obvious, exhibiting characteristics of cleavage fracture; however, the ductile fracture (B) passing through the binder is still difficult to observe. In Alloy 5, the honeycomb dimple pattern (B) is relatively enhanced, accounting for more than 20% and exhibiting an apparent ductile fracture of the binder. Thus, the fracture morphology further reflects that the toughness of the CoNiFeCr HEA binder is superior to Co.

The samples were selected to measure the crystallographic orientation and grain size by using the EBSD technique, with a and b referring to Alloy 5 and WC-Co, respectively. In order to determine the crystal orientation distribution in WC and binder, the crystallographic orientation of WC and binder was obtained with respect to the standard IPF orientation. As illustrated in Figure 9(a1,b1), for the WC grains, the red or near-red color crystals exhibit the (0001) basal plane and planes with a small angle towards the (0001) plane; the same is true for the (2-1-10) and (10-10) planes. The obtained findings demonstrate that the crystallographic orientation of the WC grains in two samples is random without any preferred orientation. Moreover, the crystallographic orientation growth of the WC grains is not affected by the CoNiFeCr HEA binder.

Figure 9(a2,b2) depicts the distribution of the WC grain size and mean free path of the binder (MFP). The average size of the WC grains in two samples is as follows: Alloy 5 (1.46 μm) < WC-Co (1.82 μm), whereas the MFP value of the samples follows the sequence: Alloy 5 (1.78 μm) > WC-Co (1.63 μm). Previous studies [20,32,33] have demonstrated that the growth of WC grains is considerably inhibited by the multi-element alloy. As illustrated in Table 3, the solubility of W in the binder of WC-Co is higher than Alloy 5, which indicates that the solubility of WC in the CoNiFeCr HEA binder is lower than that in Co. The low solubility leads to a weaker dissolution-reprecipitation effect during the sintering process, thus resulting in a small grain size. On the other hand, during the dissolution-reprecipitation process, the (Cr,W)C phase at the interface exerts a pinning force (Zener-drag) on the moving grain boundaries. During the dissolution process, the W and C atoms must first diffuse through the (Cr,W)C layer into the binder. During the re-precipitation process, the W and C atoms again need to diffuse through the (Cr, W)C layer onto a growing WC grain. Therefore, as compared to WC-Co, WC-CoNiFeCr exhibits a smaller WC grain size and a larger MFP.

As demonstrated in Figure 9(a3,b3), the grain aspect ratio of WC is closer to 1, indicating that the grain shape is close to a circle. Further, it can be observed that the mean grain aspect ratio of WC in Alloy 5 (0.52) is larger than WC-Co (0.45). In addition, the proportion of nearly round WC grains (grain aspect ratio > 0.7) in Alloy 5 is 18.58%, which is higher than the 13.93% in WC-Co. Moreover, the WC grains gradually change from columnar crystals (displayed as a quadrangle or a triangle on the two-dimensional plane) to equiaxed crystals (displayed as nearly a circle on the two-dimensional plane) as a function of the grain aspect ratio. Therefore, as compared to WC-Co, WC-CoNiFeCr exhibits a smoother WC grain shape.

### 4.4. Evolution of Mechanical Properties

In the case where the carbon content is below the lower limit, the η phase of M_6_C type is observed to appear. As the carbon content decreases, more η phases are formed. The η phase is a brittle phase with high hardness; thus, the hardness is inversely proportional to the carbon content. As observed from Table 3, in the case where the sample is in the two-phase region, the solubility of W in Co is higher than CoNiFeCr. Moreover, the micro-hardness of the binder of Alloy 5, Alloy E and WC-Co follows the sequence: 20Co (HV668) > 10Co-7Ni-2Fe-1Cr (HV591) > 10Co-8Ni-1Fe-1Cr (HV525). Meanwhile, the atomic radii of Co, Ni, Fe and Cr are similar; thus, the solid solution strengthening effect of Co-Ni-Fe-Cr is very weak. Therefore, in the two-phase region (no η phase), the hardness of the WC-Co cemented carbide is higher than that of WC-CoNiFeCr. Further, the hardness of WC-10Co-7Ni-2Fe-1Cr is higher than that of WC-10Co-8Ni-1Fe-1Cr.

Similarly, the presence of the brittle η phase significantly reduces TRS and fracture toughness. Therefore, the TRS and fracture toughness are noted to be proportional to the carbon content. In the case where the sample is in the two-phase region, the following three factors affect the TRS and fracture toughness: (1) Interface: the (Cr,W)C precipitation phase forms at the WC/HEA interface. The coherent interface between WC and (Cr,W)C with low misfit is conducive to reducing the interface energy, and the lower interface energy effectively reduces the stress concentration. (2) Binder: compared with Co, the CoNiFeCr HEA binder exhibits superior toughness. In addition, the following trend is observed: MFP of Alloy 5 (1.78 μm) > Alloy E (1.73 μm) > WC-Co (1.63 μm). A large MFP of the binder reduces the stress concentration through plastic deformation [7,34]. (3) WC grain: the (Cr,W)C phase and CoNiFeCr HEA binder inhibit the growth of the WC grains; thus, the average WC grain size in WC-CoNiFeCr is smaller than WC-Co, and the shape of the WC grains is less sharp, which releases the local stress concentration effect to a large extent [34,35]. The reduction in stress concentration makes the generation and propagation of the fatigue cracks difficult. Therefore, in the two-phase zone (no η phase), the TRS and fracture toughness of the WC-Co cemented carbide are lower than those of WC-CoNiFeCr. Further, the TRS and fracture toughness values of WC-10Co-7Ni-2Fe-1Cr are noted to be lower than those of WC-10Co-8Ni-1Fe-1Cr.

To highlight the good mechanical properties of Alloys 5 and E, we compare it with the existing WC-20 wt.% binder, illustrating the hardness, TRS and fracture toughness in Table 4 [24,29,33,36,37]. Compared with WC-Co and WC-CoNiFe without the Cr element, the hardness of WC-CoNiFeCr samples in this study is slightly lower, while the TRS and fracture toughness are increased. Moreover, the Cr/(Co+Ni+Fe+Cr) ratio was kept constant as 0.05 to avoid the precipitation of the brittle Cr-rich M_7_C_3_ carbides. Compared with the existing WC-HEA with a high Cr content, the fracture toughness is greatly improved. Therefore, a reasonable ratio of binder composition is the key to increase the toughness of the WC-CoNiFeCr cemented carbide.

## 5. Conclusions

In this study, WC-10Co-8Ni-1Fe-1Cr and WC-10Co-7Ni-2Fe-1Cr with different carbon contents were successfully prepared by using the CoNiFeCr HEA binder. The following conclusions are drawn:(1)Compared with WC-20Co (4.55%), the lower limit of carbon content in WC-10Co-8Ni-1Fe-1Cr (4.7–4.8%) and WC-10Co-7Ni-2Fe-1Cr (4.8–4.9%) is increased. With the addition of Ni, the width of the two-phase region increases, whereas the width of the two-phase region becomes narrower with the addition of Fe and Cr.(2)Due to the use of the CoNiFeCr HEA binder, the (Cr,W)C phase precipitates at the WC/HEA interface. The coherent interface with a low degree of misfit is formed between WC and (Cr,W)C, thus resulting in a significant reduction in the interface energy and stress concentration. The (Cr,W)C phase exerts a pinning force (Zener-drag) on the moving grain boundaries, which effectively inhibits the growth of the WC grains. Therefore, compared with WC-Co, the WC-CoNiFeCr exhibits a smaller WC grain size, smoother grain shape and larger mean free path (MFP) of the binder.(3)Due to the difference in the microstructure caused by the CoNiFeCr HEA binder, the hardness of WC-10Co-8Ni-1Fe-1Cr (935 HV, 3803 MPa, 21.26 MPa.m^1/2^) and WC-10Co-7Ni-2Fe-1Cr (951 HV, 3543 MPa, 20.18 MPa.m^1/2^) is slightly reduced as compared with WC-20Co (966 HV, 3146 MPa, 19.30 MPa.m^1/2^); however, the TRS and fracture toughness values are observed to be significantly improved.(4)In order to solve the reduction in hardness caused by the HEA binder, we will prepare a functionally graded cemented carbide with HEA binder to form an integrated design of high hardness and toughness.

## Figures and Tables

**Figure 1 materials-15-05780-f001:**
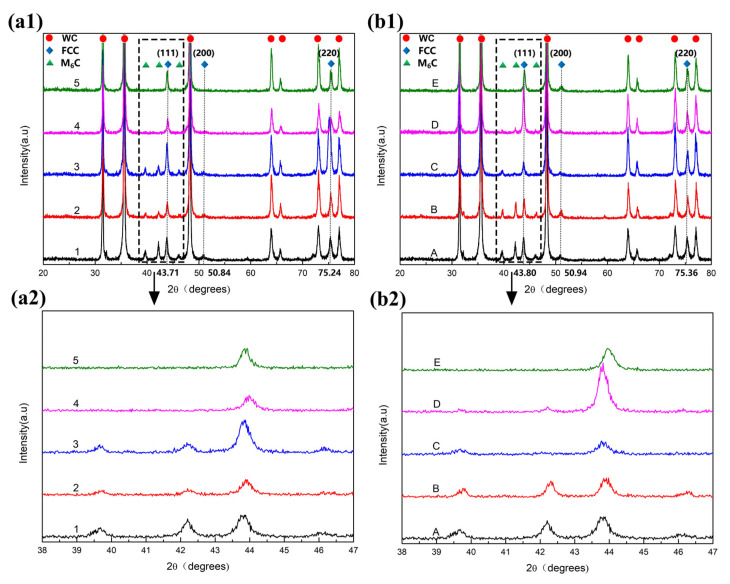
XRD patterns of Alloy 1–Alloy 5 (**a****1**,**a2**) and Alloy A–Alloy E (**b****1**,**b2**).

**Figure 2 materials-15-05780-f002:**
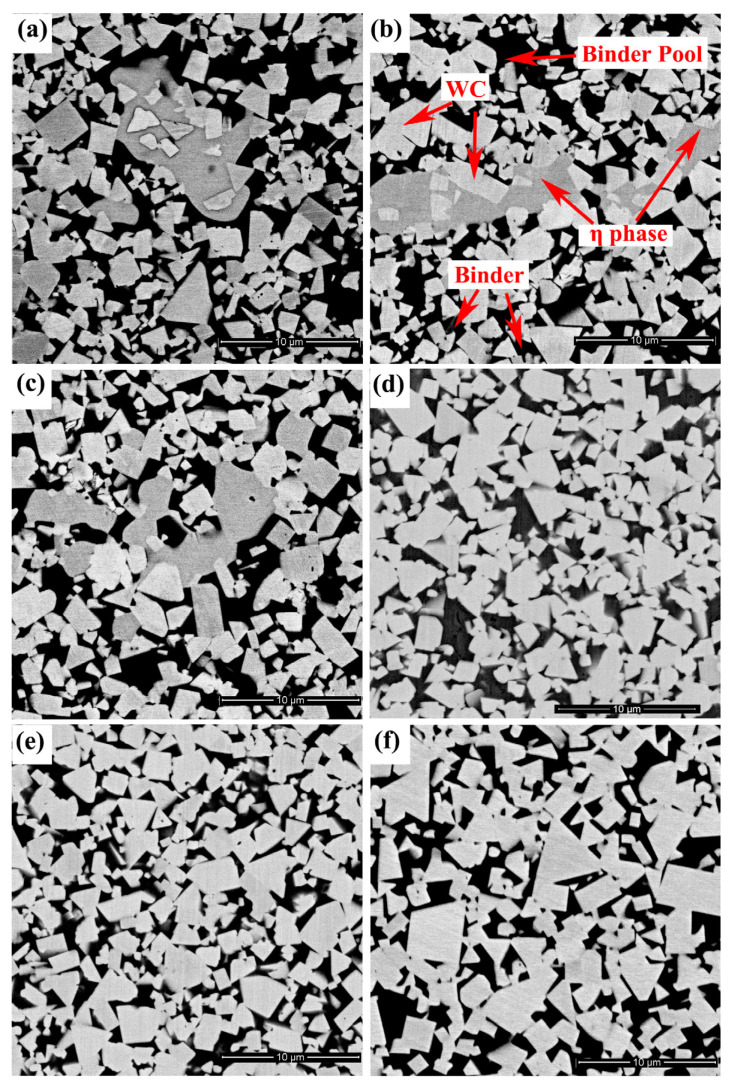
Microstructure of WC-10Co-8Ni-1Fe-1Cr: (**a**) Alloy 1, (**b**) Alloy 2, (**c**) Alloy 3, (**d**) Alloy 4 and (**e**) Alloy 5; microstructure of WC-Co (**f**).

**Figure 3 materials-15-05780-f003:**
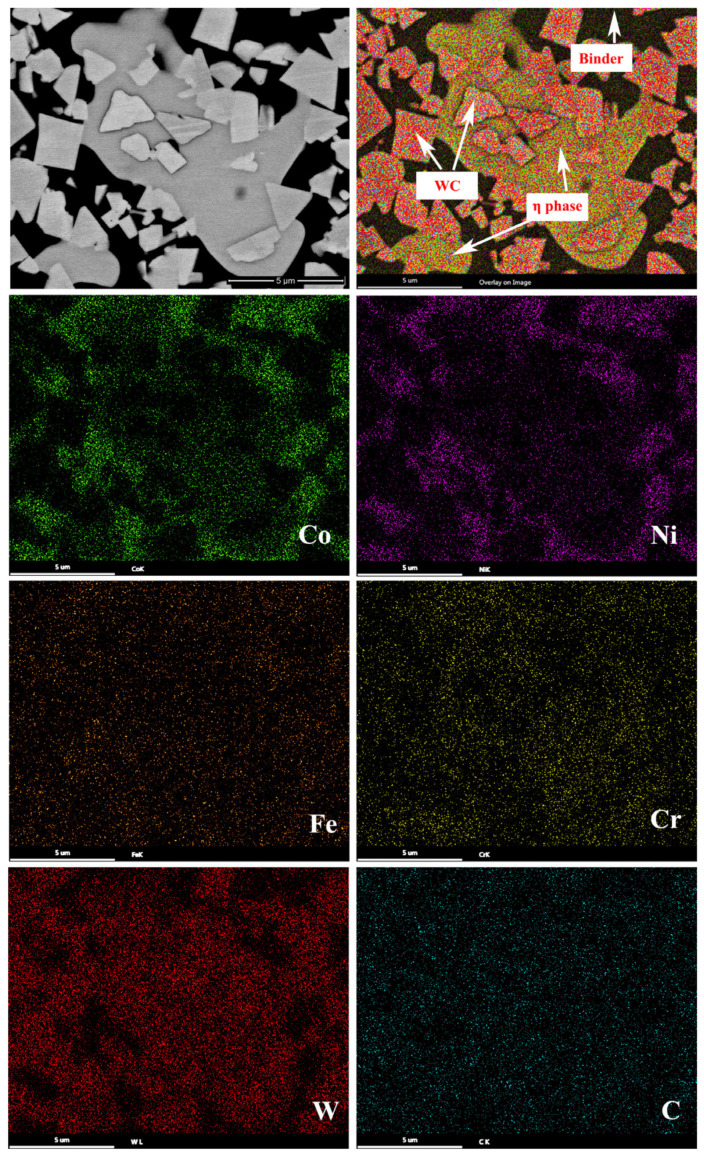
Characteristic elemental EDS-SEM mapping of Alloy 1.

**Figure 4 materials-15-05780-f004:**
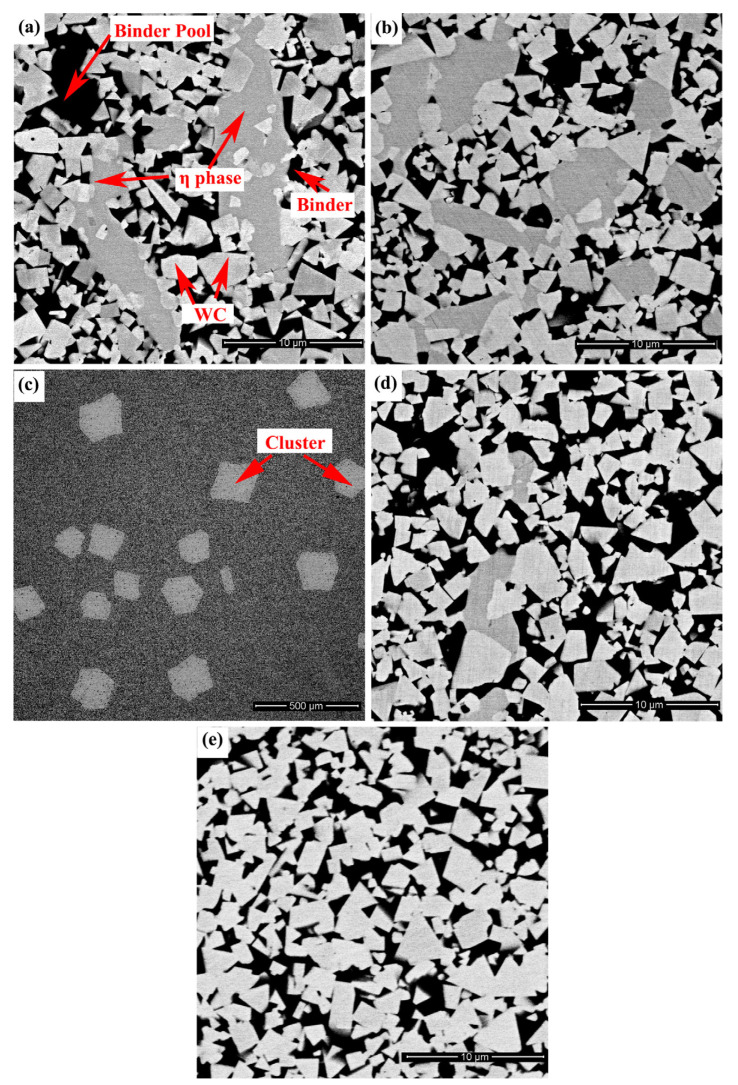
Microstructure of WC-10Co-7Ni-2Fe-1Cr: (**a**) Alloy A, (**b**) Alloy B, (**c**) Alloy C, (**d**) Alloy D and (**e**) Alloy E.

**Figure 5 materials-15-05780-f005:**
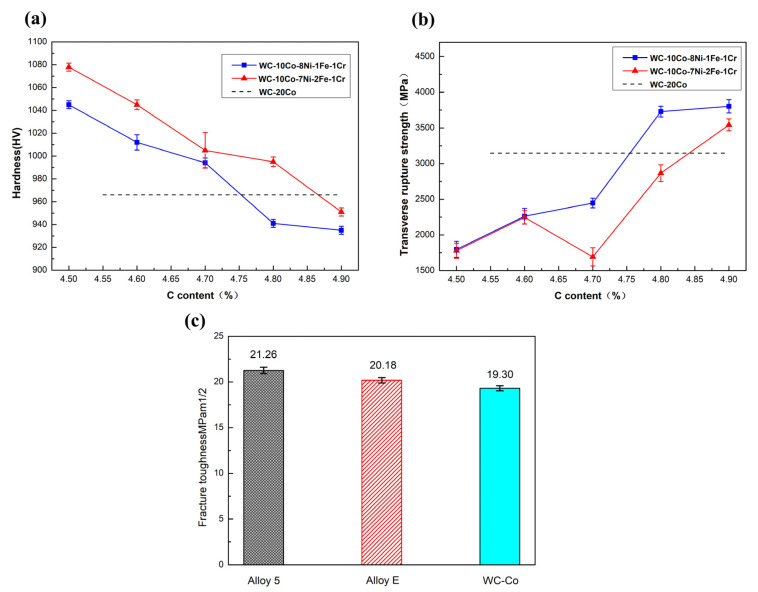
Mechanical properties of WC-CoNiFeCr and WC-Co cemented carbides as a function of the carbon content: (**a**) hardness and (**b**) TRS; (**c**) fracture toughness of Alloy 5, Alloy E and WC-Co.

**Figure 6 materials-15-05780-f006:**
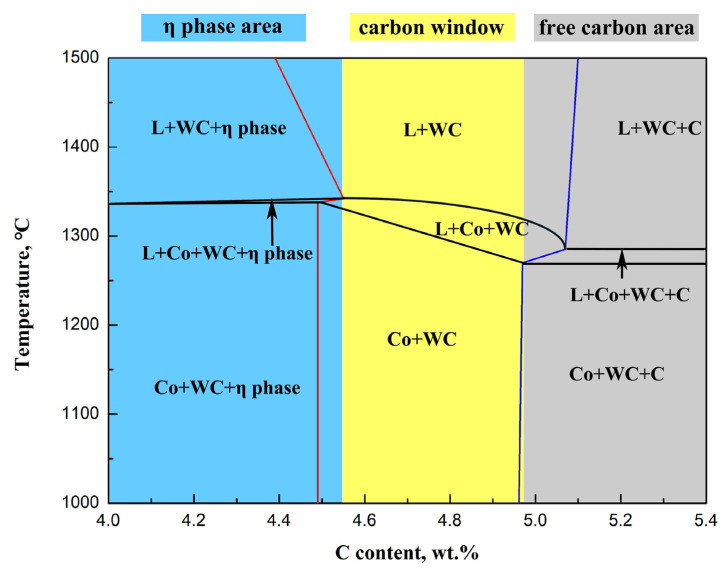
The vertical section of the phase diagram in the WC-20Co cemented carbide.

**Figure 7 materials-15-05780-f007:**
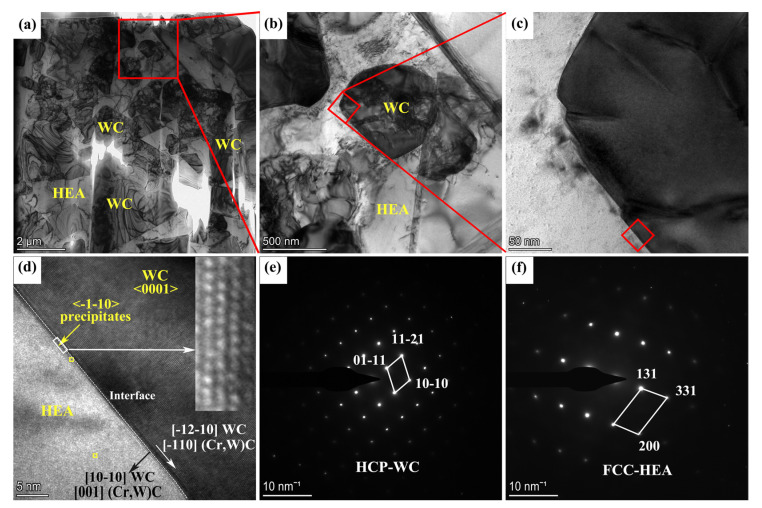
TEM analysis of Alloy 5: (**a**) TEM bright field image, (**b**) enlarged view of the red rectangle in (**a**), (**c**) steps at the WC/HEA interface, (**d**) HREM image of the WC/HEA interface viewed along <0001>_WC_//<-1-10>_(Cr,W)C_, (**e**) FFT pattern of WC and (**f**) FFT pattern of HEA.

**Figure 8 materials-15-05780-f008:**
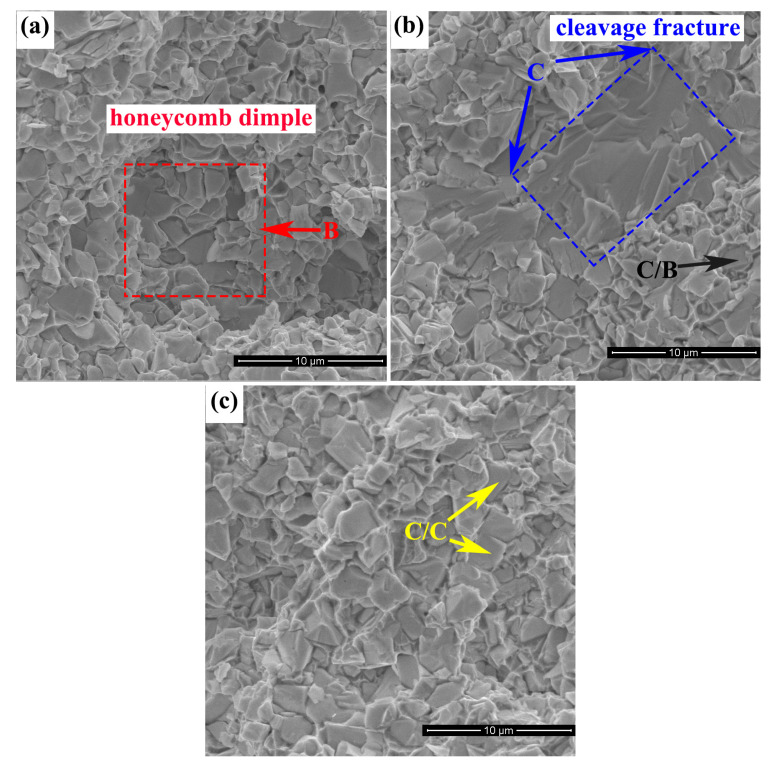
Fracture morphology of the cemented carbides: (**a**) Alloy 5, (**b**) Alloy E and (**c**) WC-Co.

**Figure 9 materials-15-05780-f009:**
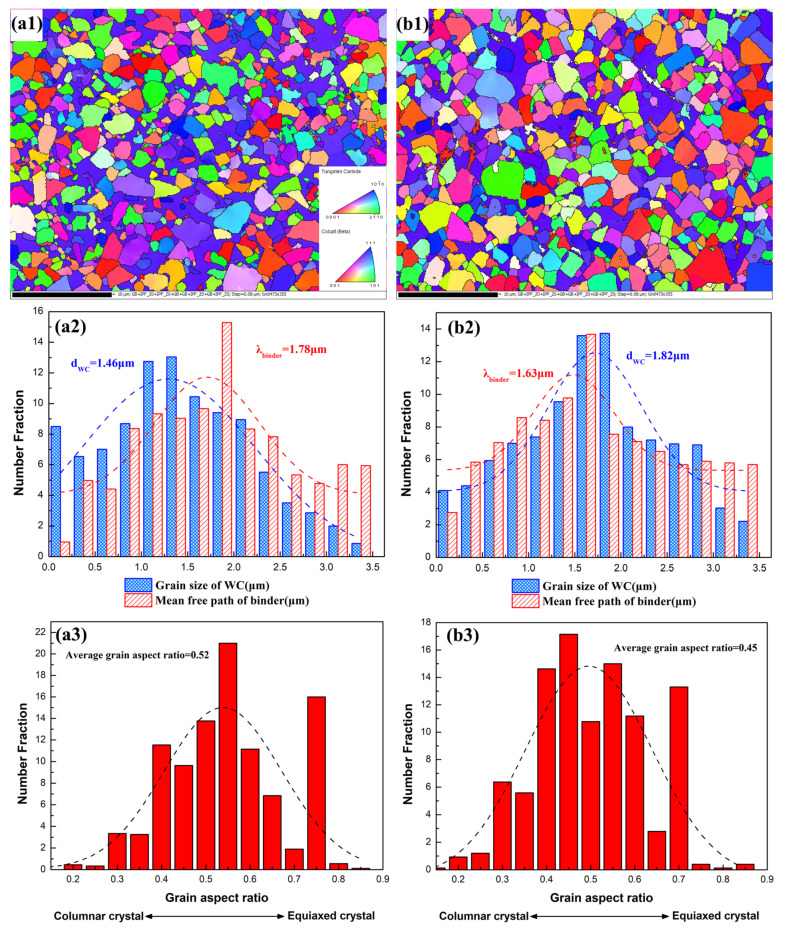
EBSD images of Alloy 5 and WC-Co, respectively (**a**,**b**): (**a1**,**b1**) inverse pole figure (IPF), (**a2**,**b2**) WC grain size and MFP distribution and (**a3**,**b3**) grain aspect ratio of WC.

**Table 1 materials-15-05780-t001:** Nominal composition, binder volume fraction (V_binder_), volume fraction of η phase (V_η_) and contiguity of the carbide phase (C_WC_) of WC-CoNiFeCr and WC-Co.

Samples	Composition/wt.%						Carbon Content in Alloy	V_binder_ (%)	V_η_ (%)	C_WC_
	WC	W	Co	Ni	Fe	Cr_2_C_3_				
Alloy 1	70.96	8.89	10	8	1	1.15	4.5%	24.18	22.1	0.283
Alloy 2	72.59	7.26	10	8	1	1.15	4.6%	25.1	13.7	0.305
Alloy 3	74.23	5.62	10	8	1	1.15	4.7%	28.2	10.3	0.307
Alloy 4	75.86	3.99	10	8	1	1.15	4.8%	31.8	-	0.335
Alloy 5	77.49	2.36	10	8	1	1.15	4.9%	32.2	-	0.333
Alloy A	70.96	8.89	10	7	2	1.15	4.5%	20.3	23.7	0.289
Alloy B	72.59	7.26	10	7	2	1.15	4.6%	23.2	15.6	0.301
Alloy C	74.23	5.62	10	7	2	1.15	4.7%	-	-	-
Alloy D	75.86	3.99	10	7	2	1.15	4.8%	27.6	11.0	0.307
Alloy E	77.49	2.36	10	7	2	1.15	4.9%	33.2	-	0.328
WC-Co	80	—	20	—	—	—	4.9%	34.6	-	0.322

**Table 2 materials-15-05780-t002:** EDS composition (at.%) in the binder and at the interface in Alloy 5.

	W	C	Co	Ni	Fe	Cr	Cr/Co+Ni+Fe+Cr
Binder	4.29	0.79	50.92	35.39	4.57	4.04	0.043
Interface	38.11	21.42	18.47	12.06	2.39	7.55	0.187

**Table 3 materials-15-05780-t003:** Binder composition in different samples.

Samples	Composition of Binder (wt.%)
Alloy 1	30Co-24Ni-3Fe-3Cr-40W
Alloy 2	34Co-29Ni-4Fe-3Cr-30W
Alloy 3	36Co-30Ni-4Fe-3Cr-27W
Alloy 4	38Co-30Ni-4Fe-4Cr-24W
Alloy 5	39Co-31Ni-4Fe-4Cr-22W
Alloy A	30Co-23Ni-7Fe-3Cr-37W
Alloy B	32Co-24Ni-7Fe-3Cr-34W
Alloy C	34Co-25Ni-7Fe-3Cr-31W
Alloy D	36Co-26Ni-7Fe-4Cr-27W
Alloy E	37Co-27Ni-8Fe-4Cr-24W
WC-Co	61Co-39W

**Table 4 materials-15-05780-t004:** The mechanical properties of the WC-20binder samples.

Samples	Hardness (HV)	TRS (MPa)	*K_Ic_* (MPam^1/2^)
Alloy 5	935	3803	21.26
Alloy E	951	3543	20.18
WC-Co	966	3146	19.30
WC-CoNiFe [29]	1087	3069	18.97
WC-Al_0.5_CrCoCuFeNi [36]	1413	-	17.4
WC-Co/CrMnFeCoNi [24]	1330	-	17.8
WC-CoCrCuFeNi (low carbon) [37]	1085	-	7.8
WC-CoCrCuFeNi (high carbon) [37]	922	-	7.6
WC-AlCrFeCoNi [33]	1600	-	9.2

## Data Availability

The study did not report any data.

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
