# Peer review of "The Effect of Carbon Content on the Microstructure and Mechanical Properties of Cemented Carbides with a CoNiFeCr High Entropy Alloy Binder"

_materials, 2022, doi:10.3390/ma15165780_

Round 1

Reviewer 1 Report

Overall, the paper is well prepared, but there are needed some improvements, in order to be published:
1.    English revision
2.    In the Introduction section, the last paragraph should contain the scientific contribution and scientific hypotheses of your research. Complete, further elaborate the scientific contribution and scientific hypotheses of your research. Be explicit. In addition to the goal of the research (which was written), the novelty in the context of the scientific contribution should be pointed out. Scientific contributions should be written based on the shortcomings of previous research in the literature. In this way, the authors will better emphasize novelty and scientific soundness.
3.    Check accuracy of data interpretation (XRD peaks unidentified). Fig 2. The grain size interpretation is subjective without data. The mechanical properties are influenced greatly by the grain size.
4.    In the conclusions, state the scientific contribution, the shortcomings of your methodology and future research. How are the proposed materials better than the classical WC-Co composition from the mechanical properties point of view. Besides the Co downsides, from discussions section this aspect it is not clear to me.
5.    Generally, the quality of the writing could be improved

Author Response

Response to Reviewer 1 Comments

Dear Editor,

Thank you for your information of reviewing of our manuscript. We have revised the manuscript according to the reviewers’ comments line by line. The revised parts are also marked in red in the manuscript.

Point 1:English revision
Response 1: English expression has been polished in the manuscript.

Point 2: In the Introduction section, the last paragraph should contain the scientific contribution and scientific hypotheses of your research. Complete, further elaborate the scientific contribution and scientific hypotheses of your research. Be explicit. In addition to the goal of the research (which was written), the novelty in the context of the scientific contribution should be pointed out. Scientific contributions should be written based on the shortcomings of previous research in the literature. In this way, the authors will better emphasize novelty and scientific soundness.

Response 2: The scientific contribution and scientific hypotheses of the research were added in the Introduction section.

Point 3: Check accuracy of data interpretation (XRD peaks unidentified). Fig 2. The grain size interpretation is subjective without data. The mechanical properties are influenced greatly by the grain size.

Response 3: The XRD data was checked carefully and the enlarged view of the small XRD peaks was added in Fig. 1. The grain size was measured by using the intercept method. When the η phase exists, the growth of WC grain is severely inhibited, so the grain sizes of η phase, binder pools and WC grains have no guiding significance for the mechanical properties of cemented carbide. When the alloy was in the two-phase region, the grain size of WC and the mean free path of the binder have been counted by EBSD in Fig. 9.

Point 4: In the conclusions, state the scientific contribution, the shortcomings of your methodology and future research. How are the proposed materials better than the classical WC-Co composition from the mechanical properties point of view. Besides the Co downsides, from discussions section this aspect it is not clear to me.

Response 4: The scientific contribution, the shortcomings of your methodology and future research were added in the conclusions. Due to the use of the HEA binder, the (Cr,W)C phase precipitated at the WC/HEA interface, and a coherent interface with a low degree of misfit was formed between WC and (Cr,W)C. The (Cr,W)C phase exerted a pinning force (Zener-drag) on the moving grain boundaries, which effectively inhibited the growth of the WC grains. Therefore, compared with WC-Co, WC-CoNiFeCr exhibits smaller WC grain size, smoother grain shape and larger mean free path (MFP) of the binder. The difference in microstructure results in higher TRS and fracture toughness.

Point 5: Generally, the quality of the writing could be improved.

Response 5: The manuscript has been refined carefully.

Should you have any questions, please feel free to contact us.

Best regards,

Huichao Cheng

Reviewer 2 Report

The authors have presented a study on the Effect of carbon content on microstructure and mechanical properties of WC-CoNiFeCr. They have examined the formed (Cr, W)C phase which precipitated at the WC/HEA interface has inhibited the growth of the WC grains resulting in slightly lower hardness with higher transverse rupture strength (TRS) and fracture toughness.  Overall, I found this article somewhat encouraging. I feel that there is very much detail in the observations, and statements are made with good supporting evidence; the article seems somewhat specific and very deep, even with the overall length of the paper. I recommend this study for publication after some minor corrections.

1.      While written English can be followed, there is certainly room for improvement. Some of the sentence construction is challenging to follow, and a few examples make very little sense.

2.      Authors should write about the full form of  “SHIP furnace”…..(some other information as well…..at least 2 lines)

3.      Nominal compositions of WC-CoNiFeCr have been reported in Table 1, Authors should also add the fraction of binder phase and contiguity value in table 1.

4.      Authors should also write the tribo-corrosion response of this composition of hardmetals.

5.      Authors should also compare the present results with the existing literature results (containing Co, Ni. HEAs binder) at the end of the diccussion part.

6.      The authors should give a good reason for the necessity of the work.

7.      Authors should add some references “Failure Behavior of Cemented Tungsten Carbide Materials: A Case Study of Mining Drill Bits” and…” A comprehensive review on synergy effect between corrosion and wear of cemented tungsten carbide tool bits: A mechanistic approach” and ”Corrosion behavior of WC-Co tool bits in simulated (concrete, soil, and mine) solutions with and without chloride additions”

Author Response

Response to Reviewer 2 Comments

Dear Editor,

Thank you for your information of reviewing of our manuscript. We have revised the manuscript according to the reviewers’ comments line by line. The revised parts are also marked in red in the manuscript.

Point 1: While written English can be followed, there is certainly room for improvement. Some of the sentence construction is challenging to follow, and a few examples make very little sense.

Response 1: English expression has been polished in the manuscript.

Point 2: Authors should write about the full form of  “SHIP furnace”…..(some other information as well…..at least 2 lines)

Response 2: The full form of  “SHIP furnace” and related information were written in Page: 3, Para: 2 and Line 4.

Point 3: Nominal compositions of WC-CoNiFeCr have been reported in Table 1, Authors should also add the fraction of binder phase and contiguity value in table 1.

Response 3: The binder volume fraction (Vbinder), the volume fraction of η phase (Vη) and the contiguity of the carbide phase (CWC) of WC-CoNiFeCr and WC-Co measured by ImageJ software from the SEM images were added in Table 1. Related descriptions were added in Page: 8, Para: 2.

Point 4: Authors should also write the tribo-corrosion response of this composition of hardmetals.

Response 4: The tribo-corrosion response of this composition of hardmetals has been tested. Due to the excessive content of this paper, it will be provided soon in the next paper.

Point 5: Authors should also compare the present results with the existing literature results (containing Co, Ni. HEAs binder) at the end of the discussion part.

Response 5: Comparison of the present results with the existing literature results of WC-Co, WC-CoNiFe and WC-HEA was added at the end of the discussion part and in Table 4.

Point 6: The authors should give a good reason for the necessity of the work.

Response 6: The reason for the necessity of the work was added in Page: 3, Para: 1.

Point 7: Authors should add some references “Failure Behavior of Cemented Tungsten Carbide Materials: A Case Study of Mining Drill Bits” and…” A comprehensive review on synergy effect between corrosion and wear of cemented tungsten carbide tool bits: A mechanistic approach” and ”Corrosion behavior of WC-Co tool bits in simulated (concrete, soil, and mine) solutions with and without chloride additions”

Response 7: References were added to the reference list.

Should you have any questions, please feel free to contact us.

Best regards,

Huichao Cheng

Round 2

Reviewer 1 Report

The authors have addresed all the mentioned aspects and the quality of the work has been improved overall.